# Complex Corrosion Properties of AISI 316L Steel Prepared by 3D Printing Technology for Possible Implant Applications

**DOI:** 10.3390/ma13071527

**Published:** 2020-03-26

**Authors:** Josef Hlinka, Martin Kraus, Jiri Hajnys, Marek Pagac, Jana Petrů, Zbigniew Brytan, Tomasz Tański

**Affiliations:** 1Department of Materials Engineering, Faculty of Materials and Technology, VSB-Technical University of Ostrava, 17. listopadu 2172/15, 708 00 Ostrava-Poruba, Czech Republic; martin.kraus@vsb.cz; 2Department of Machining, Assembly and Engineering Metrology, Faculty of Mechanical Engineering, Technical University of Ostrava, 17. listopadu 2172/15, 708 00 Ostrava-Poruba, Czech Republic; jiri.hajnys@vsb.cz (J.H.); marek.pagac@vsb.cz (M.P.); jana.petru@vsb.cz (J.P.); 3Department of Engineering Materials and Biomaterials, Faculty of Mechanical Engineering, Silesian University of Technology, Konarskiego 18a, 44-100 Gliwice, Poland; zbigniew.brytan@polsl.pl (Z.B.); Tomasz.tanski@polsl.pl (T.T.)

**Keywords:** additive manufacturing, implants, corrosion, wettability, biocompatibility, polarization, heat treatment

## Abstract

This paper deals with the investigation of complex corrosion properties of 3D printed AISI 316L steel and the influence of additional heat treatment on the resulting corrosion and mechanical parameters. There was an isotonic solution used for the simulation of the human body and a diluted sulfuric acid solution for the study of intergranular corrosion damage of the tested samples. There were significant microstructural changes found for each type of heat treatment at 650 and 1050 °C, which resulted in different corrosion properties of the tested samples. There were changes of corrosion potential, corrosion rate and polarization resistance found by the potentiodynamic polarization method. With regard to these results, the most appropriate heat treatment can be applied to applications with intended use in medicine.

## 1. Introduction

Additive manufacturing technology is a modern metallurgical method based on the principle of gradual sintering of the powder material layer by layer until the finished product is reached. With the help of selective laser melting (SLM) technology, it is possible to quickly produce fully functional and complexly-shaped parts, which are often not produced by other conventional technologies [1]. Due to the continuous development of this technology, a low porosity has been achieved, which is associated with a significant increase in the quality of manufactured parts. Nowadays, 3D printing products are broadly used in a wide range of applications, from the automotive, aerospace and aerospace industries, with a variety of tool inserts, landing gears and turbines [2]—to the medical industry where they are most often used as hard tissue replacements [3]. Nevertheless, SLM relates to high temperature gradients, which have a major influence on the resulting microstructural and mechanical properties of manufactured parts [4]. The proper choice of process parameters therefore directly affects metallurgical processes with an impact on porosity, surface character and residual stresses in the volume of the material [5]. The last of those mentioned may be eliminated by the application of heat treatment, which results in microstructural changes and can also cause the sensitization of grain boundaries or can have a significant effect on corrosion parameters, which are primarily important for the use of material within implant construction. According to ASTM standards, the material used for implant manufacturing must meet corrosion properties requirements, or else it cannot be used for this purpose [6].

Stainless steels have been commonly used in implantology for decades, especially for their corrosion resistance which is associated with self-passivation due to spontaneously-formed chromium oxides on the surface [7,8]. Corrosion characteristics of stainless steel implant surfaces can also be enhanced by chemical passivation [9], electrochemical passivation [10] or active coatings [11]. As with other materials with self-passivation ability, stainless steels are sensitive to localized forms of corrosion, especially in an environment containing highly reactive halide ions; i.e., F^−^ or Cl^−^ [12]. Pitting corrosion often occurs as a result of corrosion microcouple formation around a secondary phase particle with more noble electrochemical potential. This causes selective corrosion of a less stable surrounding metallic matrix [13]. In the case of additively manufactured stainless steel, these particles of secondary phases may originate in the impurities of powders used [14], the reaction of melted material with the atmosphere [15] or may be formed during inappropriate heat treatment [16]. As the powder’s impurities and content, and the inert atmosphere quality are process characteristics which can be easily adjusted, the effect of heat treatment on the corrosion properties of additively manufactured material plays a key role in the development process of innovative applications, from austenitic stainless steels for use in medicine to implantology. The main motivation of presented study is to compare selected properties of AISI 316L prepared by SLM and classical AISI 316L.

## 2. Material, Heat Treatment and Experimental Techniques

### 2.1. Material and Processing Parameters

Investigations were performed on the austenitic stainless steel AISI 316L prepared by the additive manufacturing process from atomized powder certified by Renishaw with an average particle size of 45 ± 15 μm. The chemical composition according to the Renishaw certificate is listed in the following Table 1.

The Renishaw AM400 device in selective laser melting mode was used for sample production. The setup parameters used for manufacturing are presented in Table 2.

After the LSM process, the samples in the shape of a longitudinally cut letter “H” were separated from the supporting plate and cleansed of powder residues using an ultrasonic bath with demineralized water and acetone for 5 minutes each. After that, the middle sections of the samples were mechanically cut off by a diamond rotary blade with a water cooling system to prevent overheating and structural changes in the material. The sample shapes with marks for cutting are illustrated in Figure 1.

### 2.2. Heat Treatment

The samples were further mechanically grinded using rotary sandpapers (grid 100–200–400–800–1500). A Struers machine was used until all SLM surface relief marks disappeared, leaving the surface with shallow scratches. At this stage, samples were divided into two categories according to the planned heat treatment. The third category contained only reference samples where no heat treatment was applied (Table 3).

The heat treatment parameters were selected intentionally according to [8,17,18,19] to simulate conditions of previously performed experiments with promising results for use in the biomechanical engineering industry. While the main reason for applying 650 °C/30 min of annealing was to reduce residual stress without significant microstructural changes, diffusion processes running during 1050 °C/30 min causes complete recrystallization with a reduction of texture and SLM artefacts in the microstructure As the cooling was very slow with a low temperature gradient, there was also no phase transformation resulting in mechanical stress accumulation expected. The standard annealing heat treatment of austenitic stainless steels applies a fast cooling rate after austenitization to prevent chromium carbide precipitation during cooling. In the present study, AISI 316L grade is not affected by any precipitation processes during the slow cooling rate applied due to low carbon content. Thus, the slow cooling can effectively eliminate residual stresses without introducing further ones during cooling. To avoid a chemical reaction between the free metallic surface and the atmosphere, the heat treatment was performed in a low pressure (1300 Pa) inert gas (argon) atmosphere. 

### 2.3. Chemical Composition, Microstructure and Metallography Observation

The purity of the manufacturing chamber atmosphere together with other parameters of the SLM process may cause minor changes to the final sample chemical composition, which may vary insignificantly from the powder used for its preparation. In particular, interstitial atoms (C,N,O) present in the air in the form of gases or organic pollutants may form secondary phases and negatively affect the surface and corrosion properties of the final material [20]. Therefore the chemical composition of the samples was verified by glow discharge optical emission spectrometry (GDOES) using a GDA 750 device from Spectruma (Hof, Germany). This method uses plasma generated by strong electric field for sputtering of the sample atoms, which are further analyzed and quantified. 

For the evaluation of the corrosive effect and the basic semi-quantitative chemical properties, an analysis of the surface layer was performed using an SEM FEI 450 Quanta FEG ( FEI Company, Brno, Czech Republic) equipped with an EDAX EDS detector (AMATEK Company, Tilburg, Netherlands) in the secondary electron mode. Accelerating the voltage to 15 keV allowed us to analyze a wide range of chemical elements from the periodic table. Due to the shape of the analyzed sample, the working distance was 11–12 mm. The metallography observations were performed on samples after mechanical polishing using equipment and diamond suspensions made by Struers (Roztoky, Czech Republic) with chemical etching (22 °C/60s) in a modified Vilella’s reagent [21] containing 10 parts 35%HCl, 10 parts distilled H_2_O and 1 part 65% HNO_3_. The image capturing and evaluation was performed by an Olympus IX70 inverted metallographic microscope (Olympus, Praque, Czech Republic)

### 2.4. Corrosion Testing Methods

AISI 316L is a material with a very low corrosion rate under normal conditions. It follows from the aforementioned material characteristics that accelerated testing methods had to be used; otherwise the standard material immersion tests could take decades. Corrosion tests were performed on a Voltalab PGZ 100 with Voltamaster 10 software (Villeurbanne, France). The test methods were selected and performed according to ASTM F 2129, ASTM G 61 and ISO 12732 with certain temperature and gas bubbling modifications in regard to subsequent application in biomedical engineering. All corrosion tests were performed in customized HDPP/HDPE corrosion cells with a lower hole exposing 0.5 cm^2^ of the tested surface. This setup allows bubbles formed on the surface during tests to escape and not to affect continuity of the measurements. A three-electrode setup, composed of a sample connected as a working electrode and a saturated calomel electrode (SCE, +241 mV vs. Saturated Hydrogen Electrode (SHE)) [22], was set as a reference electrode, and a high purity carbon rod was connected as an auxiliary electrode. The physiological saline solution (0.9 wt. % NaCl in distilled H_2_O) was used as a corrosion solution for potentiodynamic polarization, as well as open circuit potential (OCP) tests to intentionally simulate the environment of living tissue. The temperature of all tests was 25 °C. OCP measurements were performed by comparing a potential set on the working electrode and reference electrode. Therefore, all potentials in this paper are against SCE [23]. 

Before starting the potentiodynamic polarization, the initial potential value was set to −100 mV vs. the potential after stabilization of the corrosion equilibrium (OCP), with the polarization rate set to 60 mV.min^−1^ [24]. The dependence of the current flowing through the potential applied to the test sample was recorded during the measurement. The potential was gradually applied to the measured sample, which increased over time with the value of the polarization rate. After a surge in the current passing through the sample, the passive layer breakdown value was recorded, and reverse polarization was started after reaching the critical current density limit (5 × 10^−3^ A cm^−2^) In reverse polarization, the voltage was gradually decreased with the value of the polarization rate and the current passing through the sample was recorded again. If the reverse polarization current reached negative values or the potential was below the initial test value, the test was terminated. The changes in electrochemical behavior due to heat treatment were studied using the double loop electrochemical potentiokinetic reactivation (DL-EPR) method. The electrolyte prepared for this test contained 2 M H_2_SO_4_ and 0.02 M KSCN in distilled water. The tests were carried out at room temperature (25 °C). 

### 2.5. Wettability and Surface Energy

The wettability of the sample was evaluated by the sessile drop method. The surface contact angle was found by the SEE system and free surface energy was calculated by Advex Instrument software. There were 2 µL droplets of double distilled water attached to the tested surface and the contact angle θ was determined by the tangent to the drop profile at the point of contact of the three phases (liquid, solid, gas) with the plane of the sample surface [25]. The free surface energy of the solid sample is determined Young’s Equation (1), where γ_S_, γ_SL_, and γ_L_ represent the interfacial tensions per unit length of the solid-vapor, solid-liquid, and liquid-vapor contact line respectively [26].
(1)$$\gamma_{sv}-\gamma_{sl}=\gamma_{lv}\;cos\;\theta$$

## 3. Results

### 3.1. Porosity

The thresholding method was used to determine porosity. Images were transformed into B/W and total percentage of black dots representing pores was calculated [27]. This method is similar to ASTM E1245. Low magnification was used to eliminate the risk of variating the pores’ concentration effects in different sample parts. Three images of polished cross-sections cuts without etching were taken for each sample at 20× magnification. The representative images of samples from each batch are shown in Figure 2. The average porosity values calculated from and are listed in Table 4. 

All samples show a similar level of porosity. It is an undeniable fact that porosity is created exclusively during the stage of the material production process and no pores are formed during subsequent stages of heat treatment [28]. The character of the pores is analogical for all samples—there are non-melted surface powder particles visible on the sides and very bottom of the pores which indicates insufficient melting during laser beam movement. This is illustrated in Figure 3. There are also microcracks visible in the sharp edges of the pores—these may act as stress concentration zones and initiate the formation of fatigue cracks [29]. Only a very small number of pores exhibit smooth edges, and these are connected to gas being trapped in microstructure during the melting process, producing gas pockets [30]. 

### 3.2. Chemical Composition

Chemical composition was repeatedly measured on the sample surfaces using the GDOES method five times in different areas to eliminate the influence of local chemical composition deviations. The results were averaged and are presented in Table 5. According to the tests, the material of the samples fully corresponds with Renishaw certification and ASTM A276-98 standard. 

### 3.3. Microstructure

Figure 4, Figure 5 and Figure 6 show images of the sample structure of HT1, HT2 and REF at 100× magnification. The images were taken in the direction of the application of individual layers of additive production and in the direction perpendicular to the direction of the application of layers (cross-section of the samples). There are melt pools clearly visible in the cross-section of the HT1 and REF samples where equiaxial austenitic grains are formed randomly in the microstructure and lie within the melt pools and across the melt pool boundaries [31].

In the longitudinal direction (direction of the application of individual layers), the relief of the individual melt pools welded together is clearly visible on the metallographic samples of HT1 and REF. The randomly formed austenitic grains lie again either within individual melt pools or across melt pool boundaries. This effect is caused by epitaxy in combination with preferable heterogeneous solidification of liquid metal on the solid edges of the melt pools where the atoms of solidifying metal take over the orientation of surrounding grains and grow preferably in line with a negative temperature gradient. No characteristics of 3D printing are visible on the HT2 sample. The structure consists of equiaxial austenitic grains. The loss of the characteristic structure is caused by a heat treatment at 1050 °C. At this temperature, a total recrystallization of the microstructure occurred. There are no signs of melt pools visible, either in a perpendicular direction or in a parallel one, with regard to layer application. The HT2 sample shows a significantly larger grain size than HT1 and REF. This is caused by coarsening accelerated by high temperature exposure [32]. 

### 3.4. Open Circuit Potential and Cyclic Polarization

Open circuit potential measurement vs. SCE mV was first performed after 1 hour from filling the corrosion cells with a physiological solution—this time gap was to allow electrochemical processes to establish an equilibrium between oxidation and reduction based sub reactions [33]. Following OCP, measurements were performed with 24-hour periodicity. During this period, approximately one half of the corrosion solution volume was replaced by fresh solution after each measurement to avoid the risk of bacterial colonies forming in the solution and on the corrosion cell walls, which could affect solution characteristics and the results themselves, respectively. After 169 h (seven days), the last values of OCP were measured. The results of the measurements are shown in Table 6 and then graphically depicted in the chart in Figure 7, where the individual points are fitted with a suitable trend of second grade polynomic curve and the evolution of OCP in time can be evaluated for each sample

From the very negative values of OCP measured after 1 h of sample exposition, it can be concluded that the surfaces of all samples were actively oxidized and only thin, spontaneously-formed oxide layers were covering the exposed surface [34]. With an increase of exposition time, the OCP potentials of all samples shifted to more positive values; this is related to the formation of a more electrochemically stable corrosion product on the exposed surfaces, probably in the form of hydrated oxides [35] and hydroxides [36]. The equilibria of the reactions occurring on exposed surfaces are then shifted to the side of the reduction processes which results in more noble potentials measured after more extended times of exposition of all samples. At the end of the testing procedure, i.e., after 169 hours, potentials of all samples were significantly elevated and became more noble, with the deviations of OCP between starting and final measured values being more significant for heat treated samples. In contrast, however, the reference sample showed a more respectable OCP evolution in time. After OCP testing procedure was finalized for all samples, the potentiodynamic polarization test was started. Since the previous procedure was completely non-invasive, there is no risk of results being affected by previous testing. The changes of OCP were determinated by electrochemical processes occurring naturally on the tested surface, not by the testing method itself. On the other hand, the potentiodynamic polarization test is a very invasive procedure in which the surface is actively corroded due to nature of polarization procedure. Therefore, the test cannot be performed twice in the same area [9]. When potentiodynamic polarization tests of all samples after 169 hours of exposure were finished, the samples were removed from the corrosion cell, and then re-mounted and slightly shifted from previous positions. Hence, the area for new tests was unaffected and completely intact. The corrosion cell was again filled with fresh corrosion solution and a one-hour delay was applied before performing the next potentiodynamic tests. In the end, there were two polarization curves for each sample measured (after 1 and 169 hours of exposition), which can be used to determine corrosion behavior evolution in time. The corrosion curves measured for each sample are shown in Figure 8, Figure 9 and Figure 10; the different corrosion behaviors for samples HT1, HT2 and REF after 1 h and 169 h exposition are illustrated respectively. The polarization direction is indicated by black arrows in the plots. There were corrosion potentials found, in addition to polarization resistance and a corrosion rate calculated from the initial part of the polarization curves with the characteristic “V-shape” by Tafel extrapolation [37]. There was an exchange of two electrons (Fe^0^→Fe^2+^), and an average material molar mass of 56.2 g/mol [38] was considered by calculating the corrosion rate. For the control, the Stern–Geary relation was used for determination of corrosion potential and polarization resistance [39]. These calculations were done automatically by Voltamaster 10 software and the results are listed in Table 7. The results of both methods should be comparable.

According to Tafel extrapolation, the HT2 sample shows the highest values of corrosion rate, those being 1967 nm/y and 1565 nm/y for the measurements after 1 and 169 h exposition respectively. On the contrary, the REF sample shows the most promising values of corrosion rate: measurement after 1 h showed 1234 nm/y, but this number was reduced more than seven-times to 165 nm/y for the measurement after 169 h. There was also a shift of corrosion potentials to more positive values observed for HT2 and REF samples, whereas the corrosion potential of sample HT1 was 15 mV less noble after 169 h exposition than after 1 h exposition in a physiological solution. There were also changes found in polarization resistance, which may be in correlation with formation or dissolving of the oxide layer on the top of exposed surfaces during exposition time [40]. The most significant change was again found for the REF sample, which showed a polarization rate more than four times higher at the end of exposition than in its beginning.

The surfaces of samples were studied by scanning electron microscopy after corrosion test finalization. Surprisingly, there were only very few signs of pitting corrosion found on the studied surface (Figure 11A). The surfaces of the corrosion pit bottoms were rough, which is a typical sign of accelerated anodic dissolution of metallic material from the structure. Semiquantitative EDX analysis of the pits confirmed the presence of corrosion products and residues of corrosion solution. The area of corroded pit was analyzed twice for 30s time period. The averaged results of the EDX analysis from Figure 11A (area bordered by a red square) are presented in Table 8. Most of the exposed surfaces were affected by general corrosion with an enormous sign of selective dissolution from the less stable metallic phases [41], leaving honeycomb-like structures on the exposed surfaces (Figure 11B). The fast cooling rate in SLM technology results in chemical composition microsegregation on the sub-grain level. The typical cellular-columnar morphologies of fast cooled austenitic stainless steels will corrode predominantly in less alloyed grain regions, typically in their centers. Hence, the grain boundaries enriched in alloying elements due to their segregation will slowly corrode, leaving a typical (honeycomb-like) surface relief.

### 3.5. Double Loop Electrochemical Potentiokinetic Reactivation

Electrochemical double-loop potentiokinetic assay (EPR-DL) was used to assess resistance to intergranular corrosion depending on the specified type of heat treatment. The tests were carried out on a sample surface which was oriented perpendicular to the direction of growth of the individual layers of the material during its additive manufacturing. The test consisted of two separated steps: Initial polarization at −700 mV (SCE) for 120 s to activate the tested surface.Potentiokinetic test from –350 mV to +500 mV and back with polarization rate of 4 mV.s^−1^ in both directions with triple repetition

Figure 12, Figure 13 and Figure 14 show the graphs of each individual EPR-DL measurement with the indicated direction of the polarization. The maximum corrosion current density values and potentials vs. SCE are presented in the graphs for both the activation loop and the reactivation loops. The maximal current densities from the activation and reactivation loop were compared for each sample. With the increasing values of density fractions, the influence on grain boundaries also increases and may result in intergranular corrosion [42]. The results of EPRD-DL tests are presented in Table 9. According to the results, none of the tested samples showed any grain boundary sensitization [43].

### 3.6. Surface Wettability

The surfaces for contact angle measurements were finely polished to avoid any results being affected by roughness or surface unevenness [44]. Before the test, all samples were cleaned separately in an ultrasonic acetone bath with testing surfaces facing up to avoid being scratched. The droplets were exclusively placed out of visible pores, which could affect their shape [45]. Only the REF sample was compared with a traditionally manufactured (wrought) AISI 316L steel during this test. Average results of this test with their standard deviations are presented in Table 10. Representative images of droplets on tested surfaces are shown in Figure 15A (additively manufactured) and Figure 15B (wrought). The REF sample shows significantly higher surface wettability than in the traditionally manufactured sample with the same surface character. That also resulted in a different value of surface energy.

## 4. Discussion

Samples were made using a Renishaw AM 400 with the following manufacturing process parameters: laser power 200 W, scanning speed 650 mm.s^−1^, exposure time 80 μ, laser beam diameter 80 μm, powder coating thickness 50 μm and a chessboard scanning strategy. This setup showed promising results for the production of larger samples with a reduced level of internal residual stresses [46]. Altogether, with an enhanced level of fatigue resistance [47] of AISI 316L prepared by SLM, this setup it makes this process suitable for large medical and implantology equipment. 

After SLM production, the samples were divided into three groups (HT1, HT2, REF) according to their planned heat treatment. The HT1 and HT2 samples were heat treated at 650 °C and 1050 °C respectively, and the REF sample served as a reference without any further treatment. The temperatures were selected according to the results of previous research, where annealing at 650 °C resulted in residual stress reduction and partial redistribution of small particles of secondary phases in microstructure [48]. The same effect was confirmed in this investigation, and in addition, changes in the relief of the individual melt pools were detected in the microstructure. The annealing treatment was previously used in some research [48,49,50] for complete structure homogenization and residual stress reduction. Metallographic observation also confirmed complete melt pools reliefs disappearing and only the presence of equiaxial austenitic grains in material microstructure. There was also significant grain coarsening accelerated by heat temperature, as is documented in the HT2 sample microstructure, which results in a reduction of mechanical strength and notch toughness [51]. Despite the different heat treatment conditions, all samples showed a similar average level of internal porosity (0.03–0.08 %), which is significantly lower than for classical casting or powder metallurgy techniques. [52]. Those pores mostly originated in improperly melted areas as their volume was filled by intact round particles of raw powder. The inner pore walls profile shows a dendritic structure, which indicates quick heat transfer and oriented crystallization [53]. However, some studies informed us about a significantly elevated level of porosity [54,55,56]; a similar level of porosity was observed for AISI 316L after different post-processing procedures [57,58]. Increased porosity level may cause reduction of mechanical strength and fatigue resistance, as cracks were observed near pore edges can act as a stress concentrator [20,59]. The porosity level can be effectively reduced by hot isostatic pressing (HIP). This technique shows promising results, especially if applied to materials with empty pores; e.g., gas bubbles [60]. However, its application for materials with pores filled by intact particles is complicated and should be further studied, as the powder particles become densely packed rather than solidly bonded.

Corrosion tests were always performed in a direction perpendicular to the direction of the growth of the material. Open circuit potential was repeatedly measured in saline solution over 169 hours in 24 hour periods. The OCP of all samples changed during exposition time and became more noble. The change was most significant for the HT2 sample and least significant for the reference sample with no heat treatment. This change is related to self-formed layers of corrosion products with more noble electrochemical potential [34]. This effect was previously observed and described for the long term exposure of AISI 316L steel in chlorine ion-rich environments [61].

Based on potentiodynamic polarization measurements, the corrosion rates of all three samples were determined by the Tafel extrapolation method. The corrosion rate of the HT1 sample was found to be the lowest and most stable, as its value increased from 675 nm/year after 1 hour immersion to 790 nm/year after 169 h. The TZ2 sample showed a favorable decrease in the corrosion rate from 1967 nm/year after 1 hour to 1565 nm/year after 169 h. However, the most significant decrease in the corrosion rate was observed in the REF sample, where the decrease was from 1234 to 165 nm/year. The changes in corrosion rate are caused by continuous activation and deactivation of areas affected by chloride ion activity. There are other reasons for more aggressive behavior, reflected in the reduction in the passive region in saline solution, which can be due to chloride ions with higher charge density and higher capacity to form soluble species. By entering into the lattice film, the chloride introduces lattice defects, which reduce the resistance of the oxide film to corrosion [62]. Chlorine ions cause local depassivation of spontaneously formed protective oxide layers, and due to the presence of more noble secondary phase particles on exposed surfaces, localized corrosion may occur [63]. The registered values are far under 20 µm/year, which is the maximum corrosion rate commonly accepted for biomaterial design and application [64]. Previous studies of wrought austenitic stainless steels in chlorine ion-rich environments also showed much higher corrosion rates, 7–9 µm/y for AISI 316L [62]; 4–9 µm/y for AISI 304L [9] and 10–15 µm/y for AISI 316L at higher temperature [65]. Moreover, the corrosion rate can even be effectively decreased by chemical or electrochemical passivation in highly oxidizing environments [9]. Changes of polarization resistance of all samples are related to changes of corrosion product layer thickness in combination with passive layer stability. Increasing values of polarization resistance indicate the formation of the thicker passive layer during exposition or the accumulation of dielectric corrosion products on tested surfaces [66]. The signs of corrosion attack on the surfaces of SLM materials are completely different from standard corrosion pits typically found in austenitic stainless steels after polarization tests. In all reported experiments, e.g., [7,9,67,68,69], where corrosion current density reached a limit of 1·10^−3^ A/cm^2^, corrosion pits were formed, and base material was actively corroded. SEM observations of surfaces tested within this paper showed only a limited amount of corrosion pits. On the other hand, signs of selective corrosion attacks were documented leaving cellular-like structures of more electrochemically stable regions on tested surfaces. Such cell formations are most apparent on sample surfaces heat treated at 650°C and reference samples without any heat treatment. A similar corrosion effect was observed after an aggressive surface polishing/etching process with a HNO_3_+HF solution [70]. The semilogarithmic polarization curves of samples after 1 and 169h immersion in saline physiological solution show similar shapes and are optically comparable; however, the shift of corrosion potentials in relation to applied heat treatment is clearly obvious. Although the HT2_169h curve from Figure 9 is situated to higher current density, the calculated corrosion rate is approximately 400nm/y lower. This is due to unevenly open curve in Tafel region. The curve has more W-like shape rather than V-like shape, which resulted into shifting of the intersection of anodic and cathodic tangents to higher corrosion current values. This abnormal curve shape usually occurs when there are particles of less stable secondary phases (MnS, MoS, etc.) present at immersed surface [71]. The shape of all semilogarithmic polarization curves varies—the curve of HT2 is very close to typical polarization curve of wrought AISI 316L as the critical breakdown potential can be easily found on the curve [72]. The HT1 curves are more constant with less visible breakdown potential, and the breakdown potential of the REF sample is fully undetectable from the curve shape. This directly correlates with the microstructures of all three samples. The high temperature heat treatment of HT2 caused recrystallization and massive precipitation of secondary particles. This particles acts as the cathode while rest of the material acts as the anode and starts to corrode rapidly when breakdown potential is reached. On the other hand, fast solidification of REF sample microstructure did not allow substitute elements’ diffusion and precipitation in the form of secondary phase particles. This resulted in constant dissolving of less stable regions without the formation of galvanic microcouples. This effect was previously observed for additive manufacturing processes [73].

The results of EPR-DL showed that ASIS 316L manufactured by SLM indicates minor signs of grain boundary sensitization, and it was confirmed that both heat treatment strategies used even improved intergranular corrosion resistance. Previous studies performed on continuously casted AISI 316L [74,75] revealed significant grain boundary sensitization after heat treatment at 650 °C/H_2_O due to the precipitation of chromium-rich secondary phases along the grain boundaries (mainly the M_23_C_6_ and σ-phase). The exact opposite effect was confirmed for additively manufactured material heat treated at the same temperature used for research in this paper. It could probably be caused by long holding times (30 min) and a very slow cooling rate, when chromium atoms could defund grain boundaries and equalize their deficiency [74,76]. This resulted in an increasing of chromium volume content along grain boundaries and their possibility of spontaneous repassivation, even under conditions of aggressive sulfuric acid solution used for testing [77]. The indisputable fact is that powders used for the SLM method contained considerably lower carbon content than material used for experiments presented in studies: [74,75], which resulted in a reduced amount of chromium-rich carbide precipitated along grain boundaries, causing a less significant change of chromium dissolved in solid solution. 

Surface wettability was tested by the sessile drop method, wherein the contact angle between the tested surface and a small extra pure water droplet was measured. The SLM sample with no heat treatment was used for testing and the results were compared to continuously casted AISI 316L after the same surface preparation. This test confirmed a reduction in the contact angle for the SLM sample, which was approximately two times lower than for “standard” material. Studies have shown that increasing the wetting angle of the surface in the range of 0°–106° reduces the adhesion of osteoblasts (bone cells) to the surface. Conversely, the strongest adhesion of fibroblasts (fibrous cells) to the surface was observed at a wetting angle of 60°–80° [78]. According to this study, it can be stated that SLM AISI 316L material would be more suitable for the construction of long term implants where strong bonding to hard tissues is required. On the other hand, “standard” AISI 316L may be used for the manufacturing of short term implants, as there is mostly only soft fibrillar tissue bonded to its surface [79]. Increased surface wettability is also connected to accelerated adhesion of blood proteins, which is linked with the initial stage of the tissue healing process [80]. 

## 5. Conclusions

This novel research concerns a multidisciplinary study of the corrosion properties of AISI 316L stainless steel prepared by the selective laser melting method and the effects of two different heat treatment regimes. The heat treatment of 650 °C/30 min/furnace was used for residual stress reduction and 1050 °C/30 min/furnace was applied for complete structure homogenization. The main findings regarding heat treatment and final materials properties are as follows:▪The porosities of samples were in the 0.0%–0.08% range and were not affected by heat treatment. The pores were in the form of closed holes filled with unmelted powder particles. There were microcrack formations observed near these pores.▪Original melt pool reliefs were reduced during 650 °C heat treatment and fully removed at 1050 °C. The microstructure consisted of equiaxial austenitic grains. The structure of the sample heat treated at 1050 °C showed significant grain coarsening. Grains of the non-treated sample treated at 650 °C were shown lying within the melt pools and across melt pool boundaries.▪Open circuit potentials of all samples were elevated during 169 h exposition in saline solution. The most significant shift of OCP to more noble values was noted for the sample after 1050 °C heat treatment.▪The corrosion rate obtained by potentiodynamic polarization method was deeply under the recommended limit. The reference sample demonstrated the most promising results of corrosion rate, especially after 169 h exposure. The highest values of corrosion rate were measured for the sample after 1050 °C heat treatment and after 1 h exposition in saline solution. The signs of corrosion came in the form of the selective dissolving of microstructural components, leaving cellular-like reliefs on the exposed surfaces rather than in the corrosion pits. ▪The sample without heat treatment showed very low grain boundary sensitization, which was furthermore reduced by heat treatment. This was due to long holding times and a slow cooling process.▪The sample produced by the SLM method indicated nearly doubled surface wettability compared to “standard” ASIS 316L material. This phenomenon is related to higher surface energy and will have a positive effect on biocompatibility.

According to these results, SLM stainless steel AISI 316 shows promising properties for manufacturing medical instruments or implants, preferably for short term implantations. It was proven that heat treatment of SLM samples from AISI 316 increases their corrosion rate under the conditions of the human body. According to the results from this study, high temperature heat treatment should not be used for implants with long-term applications, wherein the amount of released ions from corroded material increases with time. Results from this study should be confirmed by clinical tests before their implementation into practice by manufacturing procedures. 

## Figures and Tables

**Figure 1 materials-13-01527-f001:**
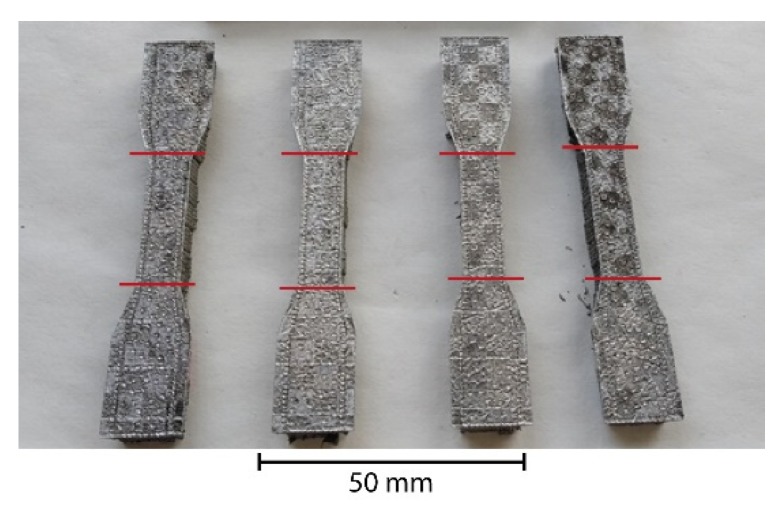
Samples for further testing.

**Figure 2 materials-13-01527-f002:**
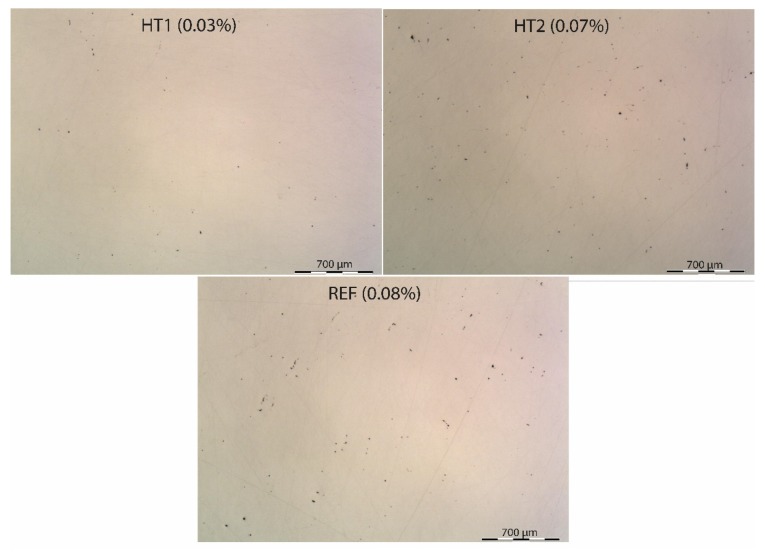
Macroimage of surfaces of HT1, HT2 and REF samples used for porosity determination.

**Figure 3 materials-13-01527-f003:**
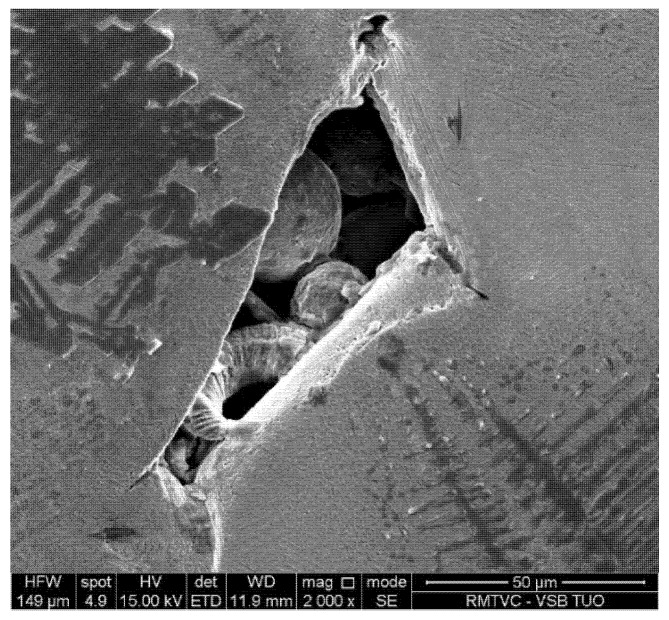
Microstructure of pores in detail, showing nonmelted round particles inside.

**Figure 4 materials-13-01527-f004:**
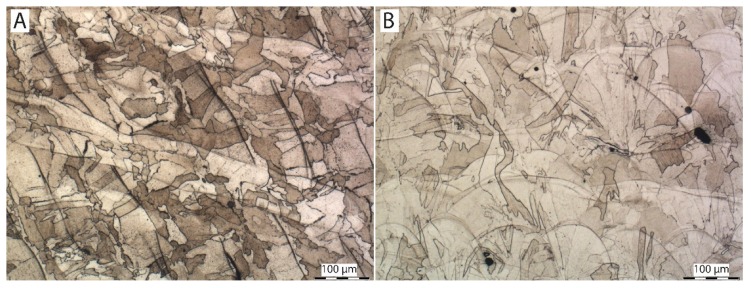
Microstructure of sample HT1: (**A**) in direction parallel with direction of application of individual layers, (**B**) in direction perpendicular to direction of application of individual layers (etched in Villella).

**Figure 5 materials-13-01527-f005:**
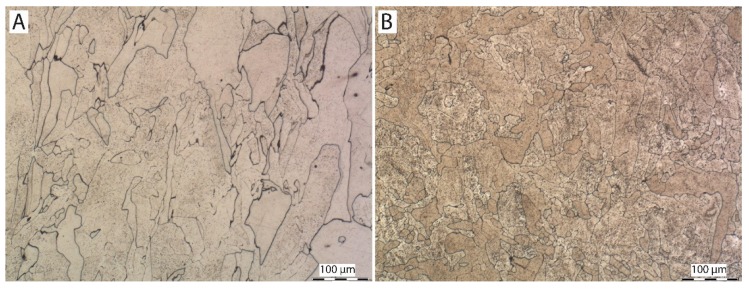
Microstructure of sample HT2: (**A**) in direction parallel with direction of application of individual layers, (**B**) in direction perpendicular to direction of application of individual layers (etched in Villella).

**Figure 6 materials-13-01527-f006:**
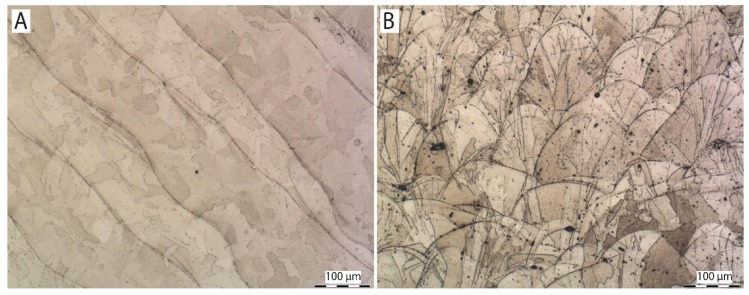
Microstructure of sample REF: (**A**) in direction parallel with direction of application of individual layers, (**B**) in direction perpendicular to direction of application of individual layers (etched in Villella).

**Figure 7 materials-13-01527-f007:**
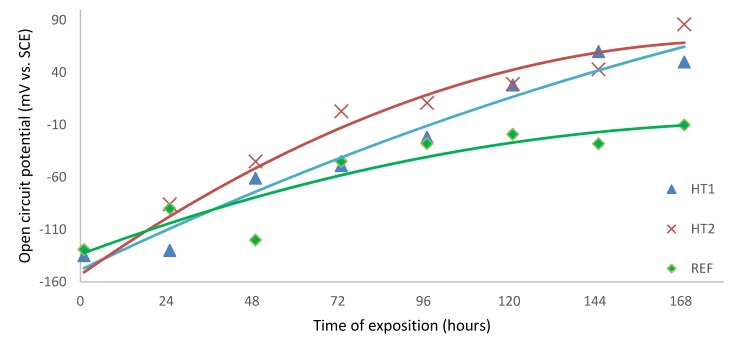
Chart of OCP evolution in 169 hours exposition for each sample.

**Figure 8 materials-13-01527-f008:**
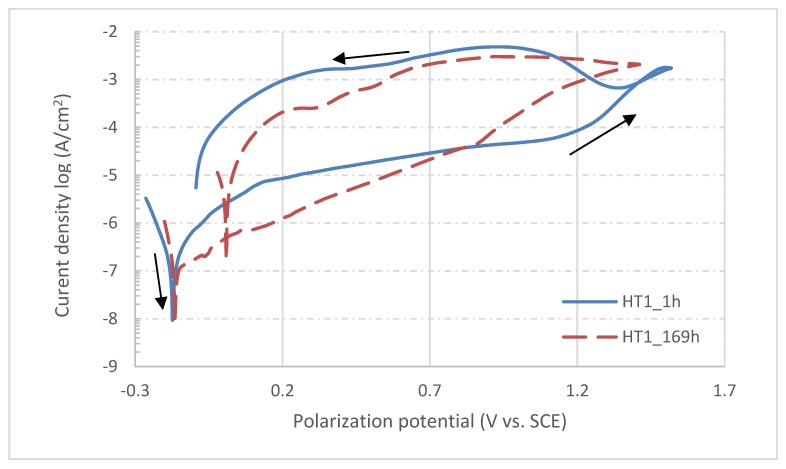
Semilogarithmic polarization curves for sample HT1 after 1 and 169 h of exposition in physiological solution.

**Figure 9 materials-13-01527-f009:**
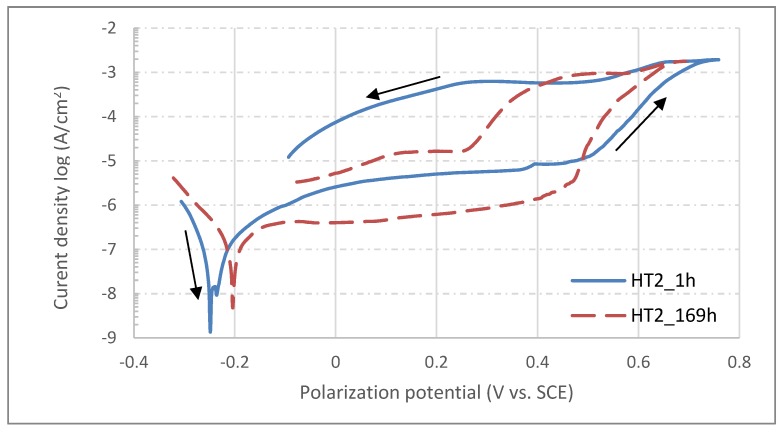
Semilogarithmic polarization curves for sample HT2 after 1 and 169 h exposition in physiological solution.

**Figure 10 materials-13-01527-f010:**
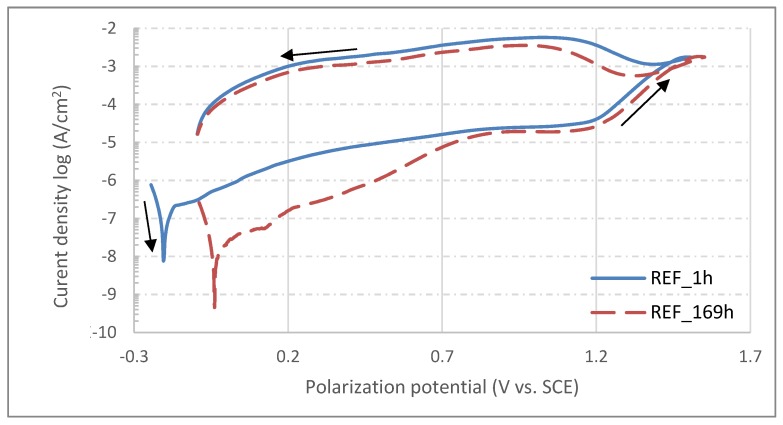
Semilogarithmic polarization curves for sample REF after 1 and 169 h exposition in physiological solution.

**Figure 11 materials-13-01527-f011:**
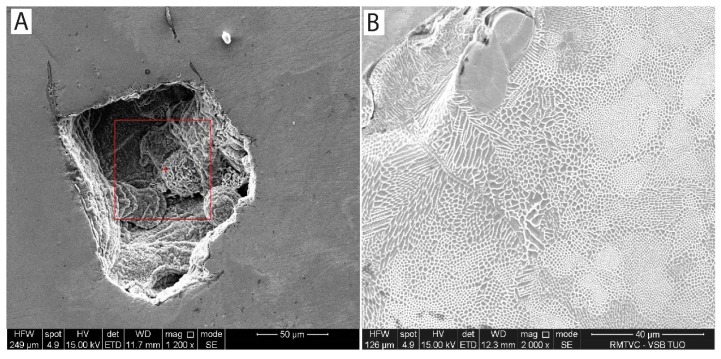
**A**-Corrosion pit with rough inner surface and traces of non-welded particles; **B**-honeycomb-like structure on the exposed surface as a sign of selective corrosion.

**Figure 12 materials-13-01527-f012:**
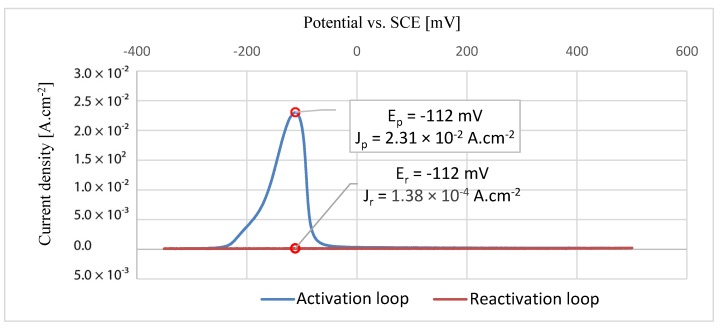
EPR-DL curve for HT1 sample.

**Figure 13 materials-13-01527-f013:**
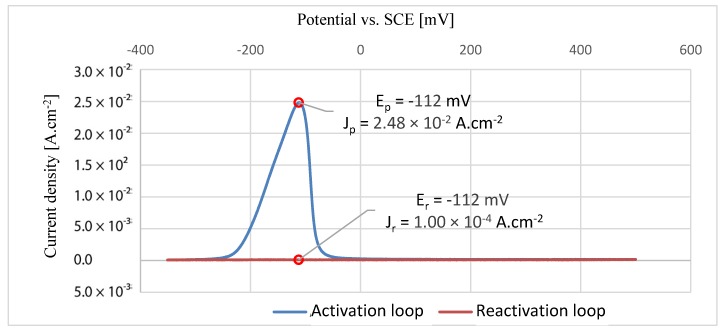
EPR-DL curve for HT2 sample.

**Figure 14 materials-13-01527-f014:**
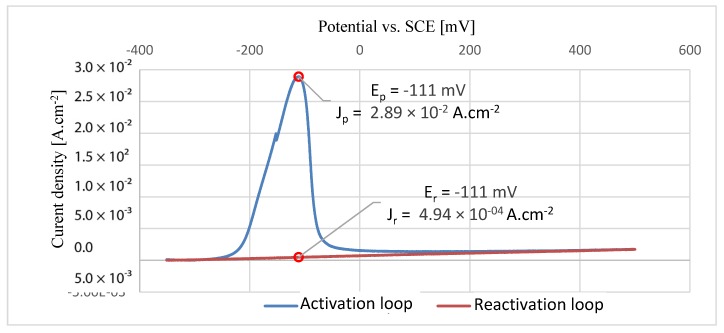
EPR-DL curve for REF sample.

**Figure 15 materials-13-01527-f015:**
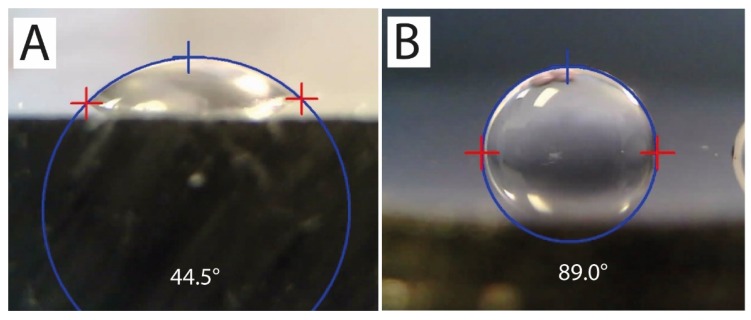
**A**-Droplet on the surface of an REF sample prepared by SLM; **B**-droplet on the surface of a wrought sample.

**Table 1 materials-13-01527-t001:** Chemical composition of atomized AISI 316L powder according to Renishaw certification.

Chemical Composition (wt. %)									
C	Si	Mn	P	S	N	Cr	Mo	Ni	Fe
Max.	Max.	Max.	Max.	Max.	Max.	Min.–Max.	Min.–Max.	Min.–Max.	Balance
0.03	1.00	2.00	0.045	0.03	0.10	16.00–18.00	2.00–3.00	10.00–14.00	

**Table 2 materials-13-01527-t002:** Parameters of SLM process.

Manufacturing Parameter	Value
Laser power (W)	200
Speed scanning (mm/s)	650
Exposure time (µs)	80
Laser beam diameter (µm)	80
Powder layer thickness (µm)	50
Hatching pattern	Chessboard

**Table 3 materials-13-01527-t003:** Parameters of heat treatment in vacuum chamber for each batch of samples.

Heat Treatment Parameters	HT1	HT2	REF (State after SLM)
Temperature	650 °C	1050 °C	-
Holding time	30 min	30 min	-
Heating rate	20 °C min^−1^	20 °C min^−1^	-
Cooling	Slow-in chamber	Slow-in chamber	-

**Table 4 materials-13-01527-t004:** Average porosity values for each sample.

Sample	Average Porosity (%)
HT1	0.03
HT2	0.07
REF	0.08

**Table 5 materials-13-01527-t005:** Averaged chemical composition obtained by GDOES method.

Chemical Composition-Content of Each Element in Tested Material AISI 316L (wt.%)								
C (%)	Mn (%)	Si (%)	P (%)	S (%)	Cr (%)	Ni (%)	Mo (%)	Cu (%)
<0.001	1.70	0.22	0.023	0.001	17.72	14.24	2.73	0.077
Co (%)	B (%)	Pb (%)	V (%)	W (%)	Al (%)	Nb (%)	Ti (%)	Fe (%)
0.048	0.0022	<0.001	<0.001	0.19	0.010	0.013	0.003	Balance

**Table 6 materials-13-01527-t006:** Open circuit potential values evolution in 169 hours of exposition.

Sample	Open Circuit Potential vs. SCE (mV)							
	1 h	25 h	49 h	73 h	97 h	121 h	145 h	169 h
HT1	−135	−130	−61	−49	−22	28	61	50
HT2	−165	−86	−45	3	11	29	43	86
REF	−129	−90	−120	−45	−28	−19	−25	−10

**Table 7 materials-13-01527-t007:** Corrosion parameters obtained by Tafel extrapolation of corrosion curves.

Sample	Corrosion Rate C_r_ (nm year^−1^)		Corrosion Potential E_cor_ (mV vs. SCE)		Polarization Resistance R_p_ (kΩ cm^2^)	
	1 h	169 h	1 h	169 h	1 h	169 h
HT1	675	790	−166	−181	110	52
HT2	1967	1565	−255	−190	102	147
REF	1234	165	−206	−47	94	408

**Table 8 materials-13-01527-t008:** Chemical composition of corrosion pit from Figure 11 A obtained by semiquantitative EDX analysis.

Chemical Composition (wt. %)						
O	Ni	Si	Cl	Cr	Mn	Fe
16.45	4.19	0.98	0.48	39.09	8.23	30.58

**Table 9 materials-13-01527-t009:** Results of EPR-DL method.

Sample	Maximal Value of Current Density for Activation Loop Jp (A.cm^−2^)	Maximal Value of Current Density for Reactivation Loop Jr (A.cm^−2^)	Current Density Fraction Jr/Jp (%)	Classification
HT1	2.31 × 10^−2^	1.38 × 10^−4^	0.60	<2%, no grain boundary sensitization
HT2	2.48 × 10^−2^	1.00 × 10^−4^	0.40	<2%, no grain boundary sensitization
REF	2.89 × 10^−2^	4.94 × 10^−4^	1.7	<2%, no grain boundary sensitization

**Table 10 materials-13-01527-t010:** Values of contact angle and calculated surface energy for REF sample and wrought sample.

Sample	Contact Angle (°)	Surface Energy (mJ.m^−2^)
REF (SLM)	43.86 ± 7.26	57.23 ± 4.45
Wrought	88.19 ± 4.99	30.37 ± 3.86

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
