# Peer review of "Complex Corrosion Properties of AISI 316L Steel Prepared by 3D Printing Technology for Possible Implant Applications"

_materials, 2020, doi:10.3390/ma13071527_

Round 1

Reviewer 1 Report

Review on Manuscript ID Materials-746471

Title: "Complex corrosion properties of AISI 316L steel prepared by 3D print technology for possible implant applications", authors: Josef Hlinka, Martin Kraus, Jiri Hajnys, Marek Pagac, Jana Petrů, Zbigniew Brytan and Tomasz Tański.

The authors study the physical properties of AISI 316L steel, and corrosion processes of these in a corrosive media represented by a sodium chloride solution.

This work is supported by experimental results and scientific research methods but the authors do not discuss the obtained results and do not compare their data with literature.

The main revisions that I consider to be made are listed below:

Major remarks

*The authors must highlight the novelty of this study by comparing their own results with results obtained in the case of 316L stainless steel used in medicine and must clearly motivate their study.

*The authors must improve the discussions on Figure 2 and Table 4.

*It is very difficult to make any comparison when using different magnitudes for all the pictures in the study (700/50/100/50/40 µm). It is advisable to use the same magnitude.

*Table 5: Fe balance?

*Figure 7 and Table 6 presents the same data and consequently I recommend to delete one of them.

*Figure 7: ”Open circuir potential (mV vs. OCP)”

The OCP is measured vs. OCP or vs. SCE?

*line 236/237: Since the previous procedure was completely non-invasive, there is no risk of results being affected by previous testing.

OCP method is a non-invasive? Since the OCP is modified with time, the method is or is not invasive?

*Lines 248-252:There were corrosion potentials found and polarization resistance and corrosion rate calculated from the initial part of polarization curves with characteristic “V like shape” by Taffel extrapolation [38]. There was exchange of 2 electrons (Fe0Fe2+) and average material molar mass 56.2 g/mol [39] considered for calculation of corrosion rate by Stern–Geary relation [40].

The explanations are so confuse! The authors used the Tafel or Stern-Geary equation to calculate the corrosion rates?

*Figures 8, 9 and 10

Potentiodynamic polarization curves (i-E) are different from semilogarithmic curves (logi-E)?! The authors represents the polarization curves or the semilogarithmic ones?

*Figures 8, 9 and 10

The authors represents the polarization curves, the semilogarithmic curves or the semilogarithms of cyclic voltammograms?

*Figure 9

According to Figure 9, the HT2_169h curve is situated to higher current density log values and consequently the corrosion rate is higher not smaller comparatively to HT2_1h curve (1967 nm/y).

*line 293: ”potentiokinetic test from –700 mV to +500 mV and back”

According to figures 12-14, the potentiokinetic test was performed from –350 mV to +500 mV

Minor remarks

line 53: ”containing active radicals, i.e. F- , Cl- ions”

Halide ions are not active radicals!? The sentence have to be modified.

Table 3: ”20 °C.min-1” should be change to ” 20 °C·min-1

Table 4: ”TZ1 and TZ2”?

What does TZ1 and TZ2 represents?

If the TZ1 reprezents HT1, the authors should use the same acronym!

Figure 7, Y-axis ”circuir” should be corrected!

Line 243: ”performin” should be corrected!

Line 328: ”into three groups (TZ1, TZ2, REF) according”

What does TZ1 and TZ2 represents?

If the TZ1 reprezents HT1, the authors should use the same acronym!

Materials journal require a specific format of references, authors must pay more attention in their writing.

There are some grammar and typing mistakes. The authors must revise the manuscript.

Author Response

Dear Reviewer.

First of all, thank you for your time spent by reviewing my paper. I've done my best with revisions. I hope the manuscript will now fully meet the journal requirements. Please find the comments to your review attached in .pdf format.

Thank you for your time,

Regards from Czech.

J. Hlinka

Reviewer 2 Report

(1) Line 79, it was said: "The samples shapes with marks for cutting are illustrated at Figure 1.4". It should be Figure 1.

(2) A scale bar should be considered to be added in Fig.1.

(3) Line 174, The Table 4 has to be rearranged.

(4) Line 185, a brief explanation of principle GDOES method should be shown in manuscript.

(5) The scale bars of Fig. 6 have to be improved. They cannot be recognized now.

(6) Line 285, Table 8, how many times did you conduct of the EDX analysis? As we know, the results of EDX can be used for qualitative analysis.

(7) Could you please support the results of hardness of three samples(HT1,HT2, REF)?

(8) A better discussion is suggested to be added for the corrosion properties of three samples. Some basic corrosion mechanisms could be referenced. In addition, the relationships between heat treatment, microstructure and corrosion behavior are important in this paper.

Author Response

Dear Reviewer.

First of all, thank you for your time spent by reviewing my paper. I've done my best with revisions. I hope the manuscript will now fully meet the journal requirements. Please find the comments to your review attached in .pdf format.

Thank you

Regards from Czech.

J. Hlinka

Round 2

Reviewer 1 Report

The authors took into consideration the observations of the referents and introduced new comments in the manuscript.

Reviewer 2 Report

The authors have responded the comments and improved the quality of manuscript. I still hold the opinion that the results of hardness should be added in this work.

(7) Could you please support the results of hardness of three samples(HT1,HT2, REF)? Sorry but this is part of another manuscript focused strictly on mechanical properties after heat treatment.